# Dye-enhanced visualization of rat whiskers for behavioral studies

**Jacopo Rigosa\*, Alessandro Lucantonio†, Giovanni Noselli†, Arash Fassihi, Erik Zorzin, Fabrizio Manzino, Francesca Pulecchi, Mathew E Diamond\***

International School for Advanced Studies, Trieste, Italy

**Abstract** Visualization and tracking of the facial whiskers is required in an increasing number of rodent studies. Although many approaches have been employed, only high-speed videography has proven adequate for measuring whisker motion and deformation during interaction with an object. However, whisker visualization and tracking is challenging for multiple reasons, primary among them the low contrast of the whisker against its background. Here, we demonstrate a fluorescent dye method suitable for visualization of one or more rat whiskers. The process makes the dyed whisker(s) easily visible against a dark background. The coloring does not influence the behavioral performance of rats trained on a vibrissal vibrotactile discrimination task, nor does it affect the whiskers' mechanical properties.

## Introduction

Many nocturnal or crepuscular rodents, including rats and mice, have long vibrissae that enable tactile exploration of the nearby environment (*Grant et al., 2009*; *Carvell and Simons, 1990*; *Gustafson and Felbain-Keramidas, 1977*; *Vincent, 1912*). Rats contact and palpate objects by projecting their facial vibrissae in front of and around the snout, giving rise to neuronal representations of what is being contacted and where it is located (*Diamond et al., 2008*; *Diamond and Arabzadeh, 2013*).

A rich variety of whisker follicle mechanoreceptors (*Ebara et al., 2002*) encode whisker deflections with remarkable sensitivity and temporal resolution (*Bale et al., 2015*; *Whiteley et al., 2015*; *Ikeda et al., 2014*). Signals follow afferent pathways to reach the primary somatosensory, secondary somatosensory and the primary motor cortex (*Ahissar et al., 2000*; *Kleinfeld et al., 2006*; *Yu et al., 2015*). Quantifying the relationship between animal's sensing of the environment and neuronal firing in these subcortical and cortical stages requires precise monitoring of whisker shape and motion during behavioral tasks.

The literature puts forward various strategies to estimate whisker movement. The electrical activity of facial muscles provides information about motor output commands, and correlates with whisking activity (*Hill et al., 2008*), but the technique includes an invasive implant and does not directly specify whisker position and deflection. Another technique is to image a high-contrast particle fixed to the whisker (*Harvey et al., 2001*; *Venkatraman et al., 2009*); however, the technique fails to track whisker shape and it likely perturbs whisker dynamics due to the particle mass and air resistance. Whisker position can be estimated by its intersection with a laser sheet (*Jadhav et al., 2009*), but this necessitates trimming the unwanted whiskers; moreover, whisker shape is lost. High-speed videography holds the greatest potential for noninvasive measurement of whisker movements, deflections, and shape (*O''Connor et al., 2010*; *Voigts et al., 2008*; *Ritt et al., 2008*; *Knutsen et al., 2005*; *Perkon et al., 2011*; *Clack et al., 2012*). Videography requires heavy computations to distinguish the target whiskers from the background, and sometimes requires trimming, which may limit behavior. Image processing for whisker tracking must be tailored to the specific

**\*For correspondence:** jacopo.
rigosa@gmail.com (JR); diamond@
sissa.it (MED)

†These authors contributed
equally to this work

**Competing interests:** The
authors declare that no
competing interests exist.

**Reviewing editor:** David
Kleinfeld, University of California,
San Diego, United States

lighting conditions. Unless the whisker is uniformly illuminated and reflects the light homogeneously (to avoid optical aberration), image quality will be degraded.

A higher degree of luminance contrast could solve many of the technical challenges inherent to videography. Identified single whiskers could be followed across sessions, and whisker-cross imaging issues could be abated. Here, we describe the application of a fluorescent dye to highlight specific sets of whiskers. The advantages of fluorescence include the conversion of each whisker into a homogeneous source of light, as well as the facilitation of multi-whisker tracking by color segmentation. However, the application of a fluorescent dye may affect the mass and the stiffness of the whisker, hence the rat's tactile perception during a behavioral task. In order to assess fluorescence as a potential tool for whisker visualization, we test performance in a vibrissal vibrotactile discrimination task (*Fassihi et al., 2014*). After finding performance unhindered, we confirm that the conserved tactile perception after application of the fluorescent dye arises from the fully conserved dynamic response of the whiskers.

## Results and discussion

### Dye-enhanced visualization

We applied two fluorescent dyes on two whiskers of a rat during brief sedation (green and red dyes - StarGazer). With a macro lens, under white light and with no optical filter, all whiskers were visible (*Figure 1A*). Despite its high quality, tracking multiple whiskers from this type of image would be challenging. Under blue light, with a long-pass filter (FEL0500 Long-pass filter cut-on wavelength 500 nm - Thorlabs) in front of the lens to remove blue reflections, the whiskers were visible in two colors and with enhanced contrast (*Figure 1B*). They would be readily separable by masking specific color ranges. Contrast can be further increased by a more red-shifted long-pass filter (red plexiglass, cut-on wavelength 625 nm measured with Implen NanoPhotometer), selecting only one dye

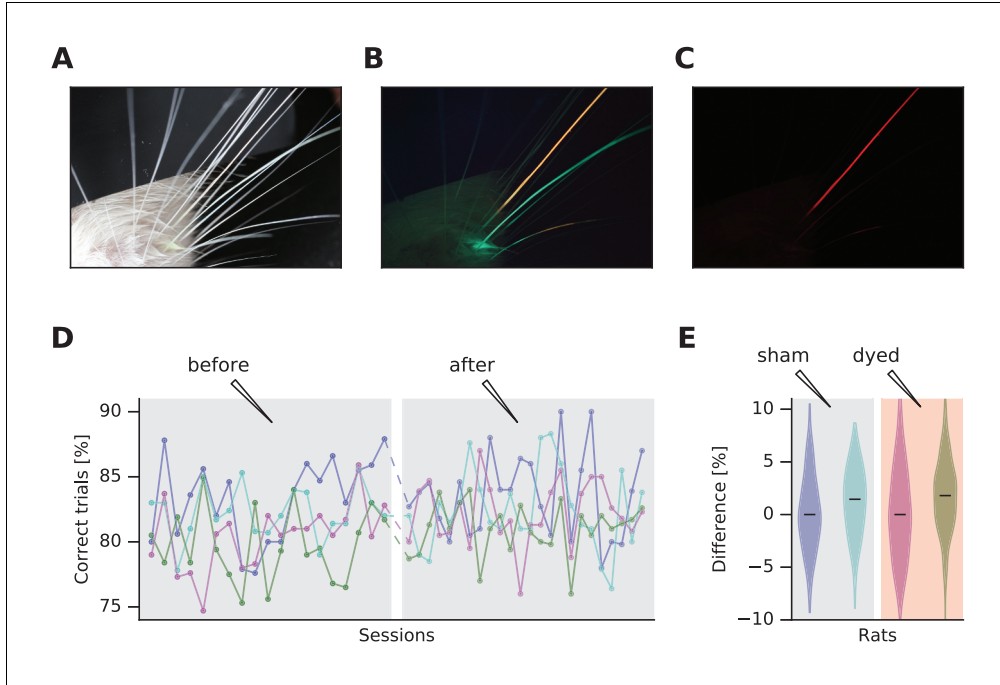

**Figure 1.** Dye-enhanced whisker visualization and effect on behavior. (**A**) Vibrissal pad with two whiskers dyed, under white light. (**B**) Same as (**A**) but under blue light and long-pass optical filter. (**C**) Same as (**B**) but with red plexiglass. (**D**) Effect of vibrissal pad coloring (pink, olive) and sham treatment (violet, jade) on daily performance of a vibrissal vibration discrimination task. Chance performance would be 50%. (**E**) Violin plots of difference in performance before and after treatment. The distributions are generated by permuting every data point before with every data point after.

(*Figure 1C*). Video recordings confirm the increased visibility of the dyed whiskers in the mobile rat (*Video 1*), facilitating tracking (*Supplementary file 1* and *Video 2*). We performed whisker tracking using JFilament (*Smith et al., 2010*), a plugin of ImageJ (*Schneider et al., 2012*), a public domain Java-based image processing program. JFilament can segment and track 2D/3D filaments and has been widely used for fluorescence microscopy images. It allows tracking of these elements through a sequence of frames and is thus suitable for whisker tracking.

### Behavioral test

We examined four rats, well-trained in a whisker-mediated vibrotactile delayed comparison task (*Fassihi et al., 2014*), for behavioral effects of dye application. Under sedation, two underwent the application of fluorescent dye while two were sham controls (saline applied). To increase the likelihood of detecting any adverse effect of dyeing, the entire vibrissal pad on the side used in the vibrotactile task was colored. The next day the rats were returned to the behavioral apparatus for testing. *Figure 1D* shows daily performance and reveals that the application of fluorescent dye, like the sham procedure, had no discernible effect on behavior. *Figure 1E* summarizes the performance, showing no difference between before versus after the dye application.

### Quantification of the frequency dynamic response of the whiskers

The behavioral tests support the notion that dye application did not affect the dynamic mechanical properties of the whiskers, but an alternative must be entertained: rats might have successfully compensated for altered whisker properties, just as they compensate for interventions as drastic as whisker clipping (*Zuo et al., 2011*). To exclude this possibility, we explored whiskers by theory – exploiting Euler-Bernoulli beam theory to perform computations by Finite Element Method (FEM) – and by experiment – measuring whisker linear dynamic response.

We simulated the frequency response function (FRF) $H$ that relates the imposed vertical displacement at the clamped base of a D2 whisker with the output displacement at any location x along its longitudinal axis (*Figure 2A*, Materials and methods). We also used high-speed videography (*Figure 2B*) to measure an extracted D2 whisker's dynamic response to white noise vibration and plotted the magnitude of the FRF (mFRF) $|H|$ along the longitudinal axis of the whisker (*Figure 2C*). The real and simulated mFRF are similar (*Figure 2A* vs *2C*), confirming previous claims that a real rat whisker behaves as a tapered beam. The mFRF constitutes a fingerprint of a whisker and thus it is a simple measure of the similarity between pairs of whiskers and the similarity of one whisker over time.

To outline the similarity index, mFRFs of whiskers designated $W_1$ and $W_2$ (similar in length, 53 and 56 mm) and $W_3$ (much shorter, 23 mm) are illustrated (*Figure 3A,B,C*, respectively). The overall similarity between any two whiskers' dynamic responses is summarized as a scatter plot (*Figure 3D*), where each point corresponds to the two whiskers' spectral values for a given [frequency, position] coordinate. In *Figure 3D*, we compare two matrices (the mFRFs) in order to test their similarity. In particular, we unwrapped these 2D matrices into two 1D vectors (i.e. we vectorized the matrices) and we computed the coefficient of determination $R^2$ of a linear regression between them. These latter values are summarized in *Figure 4* for each pair of whiskers.

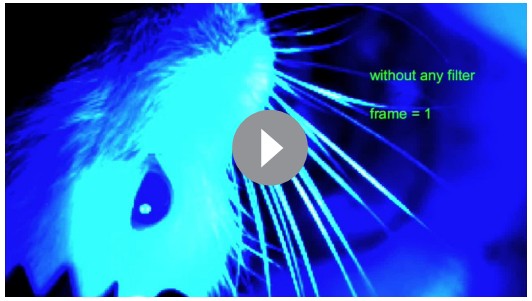

**Video 1.** Proof of concept of the method.

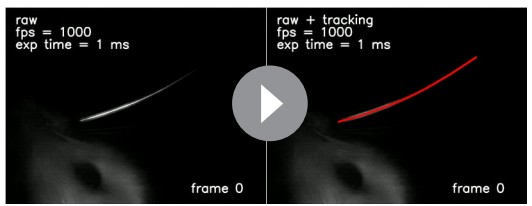

**Video 2.** Demonstration on high-speed video plus whisker tracking.

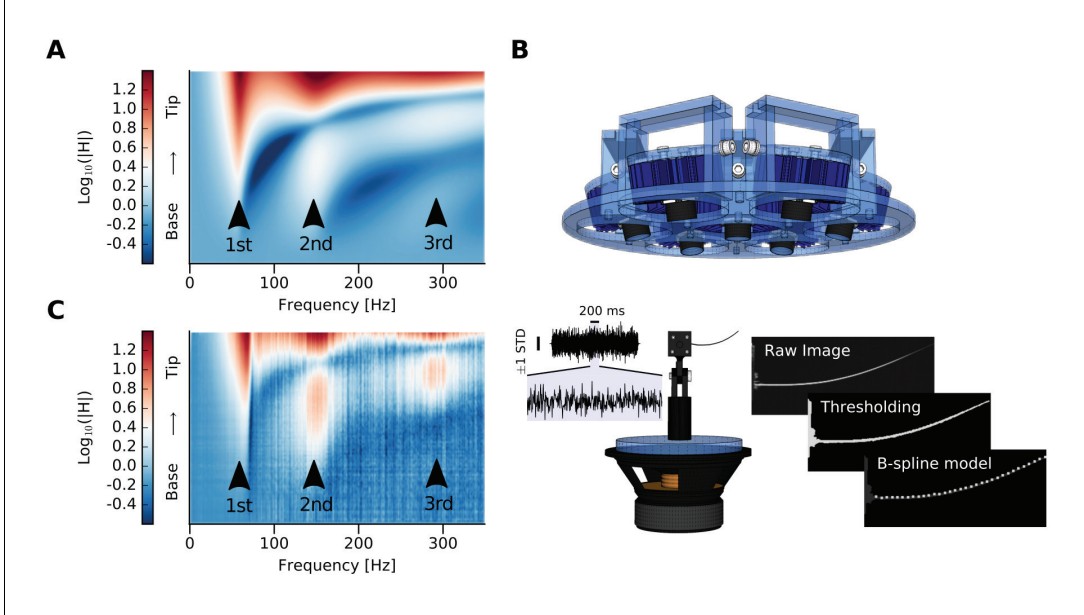

**Figure 2.** Whisker modeling and testing. (**A**) Log mFRF of the whisker approximated as a tapered beam by FEM model. Three arrowheads indicate the first three modal frequencies. (**B**) Experimental setup. The blue device at the top is the custom-made orientable lamp mounting. Below is the whisker mounted on a custom-made holder attached to the loudspeaker. To the left is a sample of the shaker input signal, which was estimated by tracking the base of the whisker and has been used as input for the computation of the mFRFs (vertical line corresponds to ± 1 standard deviation, horizontal line corresponds to 200 ms of signal, which has been also zoomed). To the right, image processing is summarized. The image is depicted from the camera's point of view, with the angle selected so as to maximize the visible motion-induced displacement. From the thresholded raw image, points belonging to the whisker are identified in the region of interest and used to compute $n$ equidistant samples along the whisker deflection using a B-spline model. (**C**) Log mFRF observed for the whisker modeled in (**A**). Three arrowheads indicate the first three modal frequencies, perfectly aligned with those of the simulation.

If two mFRFs were identical, all points would lie along the unity line and the coefficient of determination of the linear regression ($R^2$) would be 1. For $W_1$ vs $W_2$ $R^2 = 0.91$, and for $W_1$ vs $W_3$ $R^2 = 0.43$. Thus, as predicted by beam theory, a difference in whisker length corresponds to a difference in dynamic response.

## Dye effects on mechanical properties

Confident that the mFRF constitutes a robust measure of whisker dynamics, we measured a large set of extracted whiskers and quantified the similarity between any two mFRFs by the $R^2$ measure. First, we compared the dynamic responses of all whiskers, undyed, in all pairwise combinations (*Figure 4A*). The whiskers are labeled and ordered by length. Next, we compared the same set of whiskers, immediately before and after dye application (*Figure 4B*). Then, we compared the dynamic responses of all dyed whiskers, in all pairwise combinations (*Figure 4C*). The plots all attest to the stability of mFRFs in spite of dye application.

As predicted by beam theory, the mFRF is sensitive to geometry (length, cross-section area along the shaft) and to material parameters (Young's modulus, damping coefficients, mass density). We modeled the set of whiskers and computed the similarity between simulated mFRFs (*Figure 4D*). The FEM simulations (Materials and methods) uncover the same three groups of elevated mFRF similarity (orange-red blocks near the diagonal, against yellow-green-blue off diagonal) found in empirical data (*Figure 4A–C*), confirming the validity of the beam model in describing whisker dynamic response.

As a final test of stability, we examined a single whisker at many time points before and after dyeing (*Figure 4E*). Finally, to verify the sensitivity of the mFRF method to a manipulation that changes mass and stiffness, we tested the whisker after application of nail polish. The similarity matrix depicts all pairwise mFRF comparisons. Neither the passage of time nor the application of the fluorescent

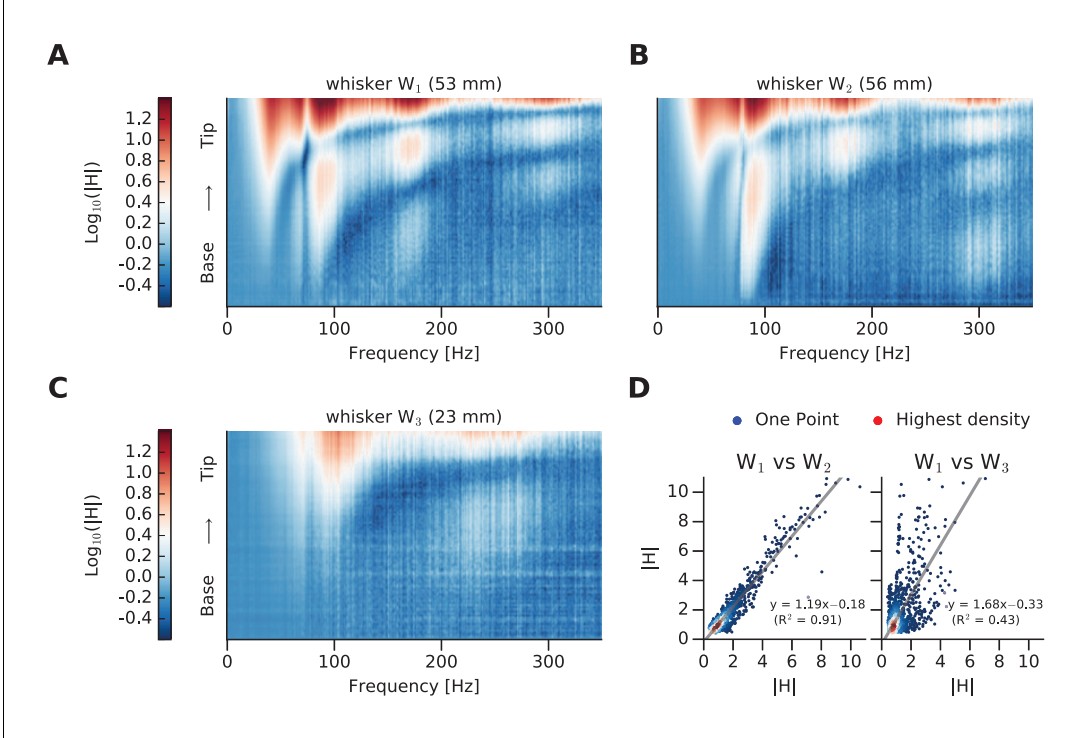

**Figure 3.** Comparison of whisker dynamic responses. (**A**) Log mFRF of whisker $W_1$. (**B**) Log mFRF of whisker $W_2$. (**C**) Log mFRF of whisker $W_3$. (**D**) Each scatter plot compares a pair of transfer functions, where each dot gives mFRF for a given frequency, position of both whiskers. On left panel $W_1$ versus $W_2$; on right panel $W_1$ versus $W_3$. The dot density has been encoded by coloring them from blue (single dot) to red.

dye had a significant effect on dynamic response ($R^2 > 0.8$ for all comparisons), whereas polish severely altered the response ($R^2 < 0.3$).

*Figure 4F* summarizes the effects of time and dye application. The upper plot gives $R^2$ values of the diagonal of *Figure 4B*, showing that for the full set of tested whiskers the similarity before/after dye application was >0.8. The lower plot gives $R^2$ values of the superdiagonal of *Figure 4E*, showing that a single whisker maintained a near-constant degree of similarity from the moment of cutting, through the dyed stage, until the application of nail polish.

In vision and audition, the external stimulus provides an adequate descriptor of the information to be processed by the sensory system. For instance, although the eyes move continuously (*Ahissar et al., 2016*), it is common to discount this motion and to consider the incoming sensory signal to be the image presented on the screen. The equivalence between external stimulus and input to the sensory processing pathway is sufficient because, to a first approximation, the eye does not change form or function as it absorbs the stimulus. The same simplification does not hold for whisker-mediated touch: the mere definition of the stimulus – a texture, a shape, a vibration – provides little information about the signal that actually engages the sensory receptors and enters the nervous system. Instead, there is a stage of pre-processing by the whisker and follicle (*Bagdasarian et al., 2013*). While the discriminandum itself can be easily quantified, the pre-processing afforded by the whisker can be difficult to measure, especially in freely moving animals.

Over the last 10 years, neuroscientists have begun to focus on sensorimotor processing in behaving rodents (*Maravall and Diamond, 2014*; *Grion et al., 2016*; *Zuo et al., 2015*). A critical step is to visualize the whiskers with clarity. The ideal methodology should be (i) simple and easy to execute, (ii) dependent on materials that are low-cost and readily available, (iii) innocuous for the animal and uninfluential on behavior, (iv) long-lasting, and (v) uninfluential on the dynamic mechanical properties of the whiskers. The method described in the present work satisfies these constraints. Its

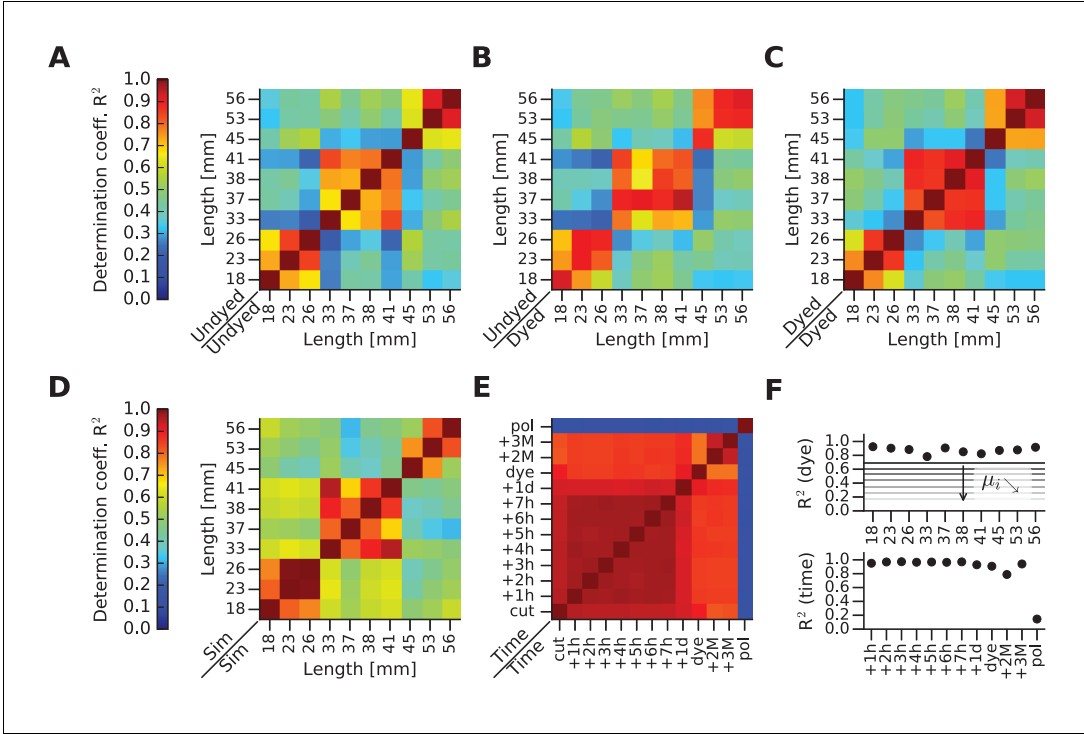

**Figure 4.** mFRF comparisons across whiskers and time. (**A**) This and the next four panels depict matrices of similarity between mFRFs in pairwise (row against column) combinations. This first matrix illustrates similarities before dye application. (**B**) Matrix comparing the dyed and undyed conditions. (**C**) Matrix comparing whiskers after dye application. (**D**) Matrix computed for FEM-simulated (Sim) whiskers. (**E**) Matrix comparing one whisker over time. Time measures begin with 'cut' (immediately after whisker extraction from the snout), 'dye' (immediately after dye application) and 'pol' (immediately after the application and drying of nail polish). (**F**) In the upper panel, the diagonal values of the matrix shown in (**B**) are plotted (each whisker against itself, before and after dyeing). The set of values obtained by taking the i-th sub- and super-diagonal values of the matrix shown in (**B**) are averaged and represented with horizontal lines labeled $\theta_i$. The further from the main diagonal, the smaller is $\theta_i$, indicating that mFRF similarity decreases with increasing disparity in whisker length. The values of the diagonal are always high, highlighting the absence of dye effect on all samples: despite the application of the dye, each whisker is always more similar to itself rather than to other whiskers. On the lower panel, the supra-diagonal values of the comparison matrix shown in (**E**), which highlights stability up to the application of polish. In particular, by reference to the horizontal lines shown in the upper panel of (**F**), the change of the whisker caused by polish can be compared to the difference between the two whiskers of length 18 mm and 56 mm.

limitation is the reduced number of vibrissae that can be simultaneously imaged and tracked. To face this challenge, future work could focus on further wavelength separation.

## Materials and methods

### Sedation for the application of the dye

Rats that underwent the application of the dye (N = 3 two were used in the behavioral study and one used to shoot the *Video 2*) and sham-control rats (N = 2) were sedated for about 2 to 2.5 hr by injection of Domitor (Medetomidine hydrochloride, dose: 0.5 mg/kg, i.p.). With the procedure terminated, an injection of Antisedan (Atipamezole hydrochloride, standard dose 0.2 ml) was used to reverse the sedated state.

### Behavioral test subjects and setup

Four male Wistar rats (Harlan Laboratories) were housed individually or with one cage mate and kept on a 14/10 light/dark cycle and reached a weight of 450–550 g. They were examined weekly by

a veterinarian. At the beginning of the experiment, they had already been trained to perform a vibrotactile delayed comparison task (*Fassihi et al., 2014*): they compared the mean speed of two vibrotactile stimuli by placing their whiskers on a $20 \times 30$ mm plate attached to the diaphragm of a shaker motor (type 4808; Bruel and Kjaer). Performance is defined as the percent of trials in which the rat correctly selected the reward side (left or right), a selection determined by the relative mean speeds of two stochastic vibrations. Double-sided adhesive was fixed to the plate to maintain the contact between the touching whiskers and the plate. Each stimulus was built by sampling a normal distribution in position centered at 0 and then applying a Butterworth low-pass filter with 150 Hz cut-off frequency. Protocols conformed to international norms and were approved by the Italian Health Ministry and the Ethics Committee of the International School for Advanced Studies.

## Collection of whisker samples

One male naive Wistar rat (Harlan Laboratories) was killed by $CO_2$ overdose in order to extract whisker samples from both vibrissal pads. To collect samples, firstly the vibrissal pad was surgically removed and then the skin was precisely cut around the whisker follicle. In this way, each sample included the follicle that mechanically wedges the hair shaft into the skin.

## Application of the fluorescent dye

We used the 'Hair Color Rinse Semi Permanent' line, in particular 'UV Red' and 'UV Green' in *Figure 1A–C*. We used 'UV Red' for the measurements of the dynamic responses (*Figures 2–4*). The ingredients, as detailed by the supplier, consist of a common base mixture to which are added specific cosmetic colorants. The common ingredients are: purified water, cetearyl alcohol, distearoyl-ethyl-hydroxyethylmonium, methosulfate, cetearyl alcohol, ceteareth-20, citric acid, and methylparaben. The specific cosmetic colorants for 'UV Red' are: CI 42520, Basic Orange 31, Basic Violet 16, Basic Yellow 40, HC Yellow 4, and CI 16255. The specific cosmetic colorants for 'UV Green' are: Basic Yellow 40, CI 42090, CI 61570, HC Yellow 4, and CI 62045.

Semi-permanent hair rinse is designed to diffuse into and bind to the hair (*Robbins, 2012*). In order to bleach the whiskers before dye application, we applied a gel of 3% hydrogen peroxide mixed with a booster powder (*Robbins, 2012*, Table 4-3). The chemical bleaching process would be expected to follow a diffusion law, where the diffusivity for the human hair has been estimated as $1.8 \times 10^{-9}$ cm$^2$/min (*Robbins, 2012*, Figure 4-3). Afterwards, we washed the whiskers with distilled water, dried them, and then submerged them for 30 min in the hair color gel (*Supplementary file 2*). The fluorescent dye method is described more in detail at Bio-protocol (*Rigosa et al., 2018*).

## Quantification of the deposited dye and of the photobleaching

As an empirical verification, we measured the mass of a set of 34 whiskers (a multiwhisker sample reduces the likelihood of measurement error) by means of a scale (RADWAG AS 220.R2) with an accuracy of 0.1 mg. Prior to processing, the pooled mass was 14.8 mg. No difference in mass was detected after the chemical bleaching, while the coloring produced an increment of 0.1 mg (from 14.8 to 14.9 mg; within the measurement error of the scale), less than 0.7%.

We measured the photobleaching of the dye for 7 hr with all seven of the lamp's LEDs pointing at the whiskers at a distance of about 15 cm. Bleaching was quantified by integrating pixel by pixel the grayscale values of an area containing colored whiskers (foreground) and another empty area of the same dimension (background). We subtracted the background from the foreground, and we normalized to the initial value in order to obtain a percentage decay (*Supplementary file 3*). We fit the data with an exponential law, the standard decay function for photobleaching. The measured time constant was about 14 hr (about 50,000 s), equivalent to about 25,000 trials of 2 s.

## Computing a mechanical fingerprint of whisker samples

As the geometry of animal whiskers resembles that of a thin, flexible filament, their dynamic response to mechanical stimuli can be suitably explored by means of beam theory (*Birdwell et al., 2007*). In particular, one can exploit the mechanical analogy between a whisker and a tapered beam (*Quist and Hartmann, 2012*; *Williams and Kramer, 2010*; *Hires et al., 2013*; *Bush et al., 2016*). A truncated conical beam model can predict rat whiskers' natural resonant frequencies (*Yan et al., 2013*), the so-called normal modes. In this study, the linear beam model of Euler-Bernoulli was

employed while accounting for whisker taper. Let $v(x,t)$ denote the beam deflection at position $x$ and time $t$. Then, by assuming a viscous damping model of the Rayleigh type, the balance laws of linear and angular momentum yield

$$A(x)\left(\frac{\partial^2 v(x,t)}{\partial t^2} + \alpha \frac{\partial v(x,t)}{\partial t}\right) + \frac{E}{\rho}\frac{\partial^2}{\partial x^2}\left(J(x)\frac{\partial^2 v(x,t)}{\partial x^2} + \beta J(x)\frac{\partial^3 v(x,t)}{\partial t \partial x^2}\right) = 0. \tag{1}$$

In the equation above, $\rho$ denotes the mass density, $A$ the cross-section area, $J$ the moment of inertia of the cross-section and $E$ the Young's modulus, whereas $\alpha$ and $\beta$ are the mass- and stiffness-proportional damping coefficients. The governing *Equation (1)* is complemented by the boundary conditions that correspond to the experimental setting (*Figure 2B*), namely

$$v(0,t) = \delta(t), \quad \frac{\partial v(0,t)}{\partial x} = 0, \quad \frac{\partial^2 v(L,t)}{\partial x^2} = 0, \quad \frac{\partial^3 v(L,t)}{\partial x^3} = 0 \tag{2}$$

for $t \in [0,\infty]$, and by the initial conditions

$$v(x,0) = 0, \quad \frac{\partial v(x,0)}{\partial t} = 0 \tag{3}$$

for $x \in [0,L]$. According to the boundary conditions of *Equation (2)*, a beam of length $L$ is clamped at its base and subject to the vertical, time-varying displacement $\delta(t)$. As for the whisker taper, we assumed the radius $r$ of the circular cross-section to linearly decrease along the whisker shaft, such that

$$r(x) = r_b - (r_b - r_t)\frac{x}{L}, \quad A(x) = \pi r(x)^2, \quad J(x) = \pi \frac{r(x)^4}{4}, \tag{4}$$

where $r_b$ and $r_t$ denote the base and tip radius, respectively.

For the computation of the beam FRF, it is now expedient to Laplace-transform the set of *Equation 1 and 2*. By taking into account the initial conditions (3), the governing *Equation (1)* simplifies into the following linear ODE in the space variable $x$

$$A(x)(s^2 + s\alpha)V(x,s) + \frac{E}{\rho}\frac{\partial^2}{\partial x^2}\left(J(x)\frac{\partial^2 V(x,s)}{\partial x^2} + s\beta J(x)\frac{\partial^2 V(x,s)}{\partial x^2}\right) = 0, \tag{5}$$

whereas the boundary conditions of *Equation (2)* yield

$$V(0,s) = \Delta(s), \quad \frac{\partial V(0,s)}{\partial x} = 0, \quad \frac{\partial^2 V(L,s)}{\partial x^2} = 0, \quad \frac{\partial^3 V(L,s)}{\partial x^3} = 0, \tag{6}$$

where $V(x,s)$ and $\Delta(s)$ denote the Laplace transform of $v(x,t)$ and $\delta(t)$, respectively. The whisker model of *Equation 5 and 6* was implemented in the commercial, finite element software COMSOL Multiphysics (version 5.2a, RRID:SCR_014767). Whiskers were accurately discretized into a finite element mesh comprising 200 elements. Cubic and quadratic shape functions were employed for the approximation of the displacement and rotation field, respectively. Since the numerical analyses were carried out in the frequency domain, the whisker FRF $H$ was readily computed by taking the ratio of $V(x,s)/\Delta(s)$ evaluated at $s = i\omega$, where $i$ is the imaginary unit and $\omega$ is the angular frequency. As for the geometric parameters of the biological samples, these were measured with the aid of custom code, which was used to process images from high-speed videos for the whisker length and bright-field micrographs for the base and tip radii. Reference material parameters were taken from the literature (*Hartmann et al., 2003*) and tuned to fit the experimental results with the numerical computations (*Supplementary file 4*).

As becomes evident from the mathematical formulation, the dynamic response of rat whiskers is sensitive to their geometric and constitutive parameters. For example, the log mFRF of a simulated D2 whisker is shown in *Figure 2A*. This reveals the characteristic peaks at the modal resonant frequencies (the first three modes are highlighted by black arrows). Modal shapes can also be inferred from the figure. Given a whisker geometry, these are distributed along the frequency axis depending upon the material parameters. Thus, a change in the modal frequencies distribution can only be explained by a change in the whisker constitutive parameters. Hence, in this work, we propose the

dynamic responses comparison as a meaningful approach to determine whether mechanical changes occurred or not as a consequence of the chemical processes needed to apply the fluorescent dye on the whisker.

## Ex vivo experimental setup and data processing

The setup (*Figure 2B*) included a mounting structure, designed to fix the sample without gluing it, a condition which would have affected the measurements. The whole system was installed on a base that was shaken by a loudspeaker (8 Ohm, 30 W - Visaton) controlled by using a custom-made Lab-VIEW code (National Instruments, RRID:SCR_014325). The set-point of the control input to the loud-speaker was computed to be a white noise sampled at 2 kHz and low-pass filtered with a Gaussian filter at 1 kHz (*Figure 2B*). The vibrational apparatus was inserted in a wooden stage completely covered with neoprene to avoid light reflections. A custom-made lamp was placed on the ceiling: it had seven bulbs (ILH-OO01-DEBL-SC211-WIR200, Wavelength 455 nm, Flux @700mA 1400 mW, Radi-ance angle ±60°; concentrator lens FA11205_Tina-D-OSL FWHM angle ±6°) that could be oriented so that the whisker sample could be homogeneously illuminated. In front of the stage, a high-speed camera (CamRecord 450, Optronis) recorded the sample vibrations at 2000 frames per second. Each frame was analyzed to compute the deflection of each whisker as a function of time (*Figure 2B*) and consequently estimate the mFRF for each trial of 2.25 s each. The mFRFs were averaged over 10 repetitions of different trials using different white noise signals, because this number of trials was sufficient in order to lower the statistic error of the measurement below the image pixel resolution.

Image processing was performed using a Python (PSF, http://www.python.org) custom code that makes use of the OpenCV library (*Bradski, 2008*). Firstly, for each frame the part of the image which contained the whisker was identified, then the points belonging to the whisker were revealed by thresholding. Hence, the whisker shape was readily estimated by means of a B-spline modeling of these points. The B-spline returns N equidistant samples along the whisker direction, which model a specific whisker segment at a specific time (*Figure 2B*). One trial thus provided N time series with a length of 4500 (2.25 s times 2000 frames per second) that portray whisker dynamics as a function of its one-dimensional space. In the linear regime, each segment of the beam is a linear time invariant system (LTI) that has an input defined as the movement of the whisker base (i.e. the segment which is closest to the loudspeaker). Hence, we estimated the mFRF for each whisker's segment using the Wiener–Khinchin theorem, which states that the autocorrelation function of a wide-sense-stationary random process has a spectral decomposition given by the power spectrum of that process and thus links the cross spectral density functions of the input and the output with the energy transfer function.

Two mFRFs correlate only if they have the same mode shape in the same modal frequencies. We define the coefficient of determination ($R^2$) of their linear regression as non-negative normalized similarity metric: $R^2$ equal to 1 corresponds to the identical responses while $R^2$ equal to 0 corresponds to completely uncorrelated responses.

## Data and code availability

The code (*Rigosa, 2017*) includes (1) the image processing from raw video and (2) the postprocessing of preprocessed data from video. A copy is archived at https://github.com/elifesciences-publications/whisker-dynamic-response. Data (*Rigosa et al., 2017*) includes one raw video to run a demo (~1 GB), the whole preprocessed dataset to reproduce results (~9 GB), a copy of the code for data acquisition and processing.

## Acknowledgements

We acknowledge the financial support of the Human Frontier Science Program (http://www. hfsp. org; project RG0015/2013), the European Research Council Advanced grants CONCEPT (http://erc. europa.eu; project 294498) and MicroMotility (project 340685), and Italian MIUR grant HANDBOT (http://hubmiur.pubblica.istruzione.it/web/ricerca/home; project GA 280778). GN gratefully acknowledges support by SISSA through the excellence program NOFYSAS 2012. The funders had no role in study design, data collection and analysis, decision to publish, or preparation of the manuscript. We also acknowledge Natalia Grion and Ben Mitchinson for their valuable consultation on whisker tracking software.

## Additional information

### Funding

| Funder | Grant reference number | Author |
| --- | --- | --- |
| Ministero dell'Istruzione, dell'Università e della Ricerca | GA 280778 | Mathew E Diamond |
| International School for Advanced Studies | NOFYSAS 2012 | Giovanni Noselli |
| Human Frontier Science Program | RG0015/2013 | Mathew E Diamond |
| European Commission | Project 294498 | Mathew E Diamond |

The funders had no role in study design, data collection and interpretation, or the decision to submit the work for publication.

### Author contributions

JR, Conceptualization, Resources, Data curation, Software, Formal analysis, Supervision, Validation, Investigation, Visualization, Methodology, Writing—original draft, Project administration, Writing—review and editing, designed the study, set up the ex vivo isolated whisker vibration apparatus, wrote data acquisition code, conducted the ex vivo whisker experiments, designed the analyses, executed the analyses and created figures, wrote the manuscript first draft, revised the manuscript; AL, Conceptualization, Resources, Software, Formal analysis, Supervision, Validation, Investigation, Visualization, Methodology, Writing—review and editing, designed the study, modeled whisker mechanical responses, designed the analyses, executed the analyses and created figures, revised the manuscript; GN, Conceptualization, Resources, Formal analysis, Supervision, Funding acquisition, Investigation, Visualization, Methodology, Writing—original draft, Project administration, Writing—review and editing, designed the study, modeled whisker mechanical responses, designed the analyses, executed the analyses and created figures, wrote the manuscript first draft, revised the manuscript; AF, Conceptualization, Resources, Supervision, Visualization, Methodology, Writing—review and editing, designed the study, set up the in vivo behavioral apparatus, conducted the behavioral experiments, designed the analyses, executed the analyses and created figures, revised the manuscript; EZ, Conceptualization, Resources, Software, Formal analysis, Supervision, Validation, Investigation, Visualization, Methodology, Writing—review and editing, designed the study, set up the ex vivo isolated whisker vibration apparatus, wrote data acquisition code, conducted the ex vivo whisker experiments, designed the analyses, executed the analyses and created figures, revised the manuscript; FM, Resources, Software, Methodology, Writing—review and editing, set up the in vivo behavioral apparatus, wrote data acquisition code, revised the manuscript; FP, Resources, Supervision, Methodology, Writing—review and editing, conducted the behavioral experiments, revised the manuscript; MED, Conceptualization, Supervision, Funding acquisition, Investigation, Visualization, Methodology, Writing—original draft, Project administration, Writing—review and editing, designed the study, designed the analyses, executed the analyses and created figures, wrote the manuscript first draft, revised the manuscript

### Author ORCIDs

Jacopo Rigosa, http://orcid.org/0000-0003-3184-994X
Arash Fassihi, http://orcid.org/0000-0001-7477-7080
Erik Zorzin, http://orcid.org/0000-0002-4083-7847
Mathew E Diamond, http://orcid.org/0000-0003-2286-4566

### Ethics

Animal experimentation: The rats were under the care of a consulting veterinarian. Protocols followed the guidelines of EU Directive 2010/63/EU, established as Italian decree 26/2014, and were approved by the SISSA Ethics Committee and the Italian Ministry of Health license numbers 569/2015-PR and 570/2015-PR.

## Additional files

### Supplementary files

• Supplementary file 1. High-speed video. A water reward spout armed with a licking sensor triggered the high-speed video recording (this is a frame of *Video 2*). The field of view is illuminated with four out of seven of the lamp's LEDs and an orange plexiglass is used as an optical filter, demonstrating a high enough signal-to-noise ratio to allow whisker tracking, which is overlaid.

• Supplementary file 2. Dye application method. After sedation with Domitor: (A) all whiskers are chemically bleached. (B) The bleaching factor is removed with wet cotton and (C) dried with airflow. (D) The single whisker is isolated by paraffin film; (E) color is applied using another paraffin film and after the application time the excess color is washed away as in (B); (F) whisker coloring is tested under the long-pass filter.

• Supplementary file 3. Photobleaching. Normalized fluorescence intensity of the whisker (dark blue); exponential decay fit (light blue); exponential time constant expressed in hours of exposition (dashed line intersection with abscissa).

• Supplementary file 4. Geometric and material parameters of the ten whiskers analyzed by the FEM model. Recall that the modal frequencies $\omega_i$ do not depend on the Young's modulus and mass density independently, as they depend on their ratio. This can be inferred from *Equation (1)* or *Equation (5)* in Materials and methods. Hence, we have tuned this ratio to fit the experimental results by keeping fixed the mass density and changing the Young's modulus.

### Major datasets

The following dataset was generated:

| Author(s) | Year | Dataset title | Dataset URL | Database, license, and accessibility information |
|---|---|---|---|---|
| Rigosa J, Lucantonio A, Noselli G, Fassihi A, Zorzin E, Manzino F, Pulecchi F, Diamond ME | 2017 | Data from: Dye-enhanced visualization of rat whiskers for behavioral studies | https://dx.doi.org/10.5061/dryad.9k9n0 | Available at Dryad Digital Repository under a CC0 Public Domain Dedication |

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
