## [Decision Letter]

Thank you for submitting your article "Dye-enhanced visualization of rat whiskers for behavioral studies" for consideration by *eLife*. Your article has been reviewed by three peer reviewers, and the evaluation has been overseen by David Kleinfeld as Reviewing Editor and Andrew King as the Senior Editor. The following individual involved in review of your submission has agreed to reveal his identity: Carl CH Petersen (Reviewer #3).

The reviewers have discussed the reviews with one another. The Reviewing Editor feels that this could be a valuable contribution to the field of vibrissa exploratory behavior as the tracking of vibrissae is essential for a large group of experimentalists and your technique offers an alternative strategy to existing bright-field and dark-field techniques. He has drafted this decision letter to help you prepare a revised submission. Since additional data are required we realize that more time may be needed, and given that this is a methods paper we will reassign this to the Tools and Resources category, as it would be inappropriate to be considered as a research Short Report.

Essential changes:

1) Few details are given about the dye and how it is applied. Exactly what dye was used, as "Stargazer" sells multiple green and red colors.

2) You need to demonstrate tracking position and shape of whiskers in the intact animal.

3) Related to point (8), accurate measurements of deflection and bending require video recording at 1 kHz or more. Is there sufficient signal-to-noise for this to occur? How much dye is deposited and what is the weight of the dye required for such a signal-to-noise ratio and how does this compare to the weight of the unpainted vibrissa? What is the intensity of the excitation light?

4) Please comment on the bleaching rate of the dye. Please comment on the ability to re-stain the vibrissae.

*Reviewer #1:*

In this technical manuscript, the authors devise a new method for whisker visualization and tracking. They use readily available fluorescent dye suitable for visualization of one or more rat whiskers. The process makes the whiskers easily visible against a dark background. The coloring does not influence the behavioral performance of rats trained on a vibrissal vibrotactile discrimination task, nor does it affect the whiskers' mechanical properties. I generally found the manuscript well organized, written in clear language and appropriately discussed in the context of the current literature, all of this making the described findings accessible to a wide readership.

Next to my overall positive impression, the paper has several drawbacks.

The authors examined four rats, well-trained in a whisker-mediated vibrotactile delayed comparison task (Fassihi 2014), for behavioral effects of dye application. However, this behavioral task does not depend on the mechanical properties of the whiskers, since whisker's movement is induced by mechanical deflection. I suggest taking these data out.

*Reviewer #2:*

Rigosa et al. offer a novel method of fluorescently dying rat whiskers in order to solve limitations of traditional high-speed video based on brightfield imaging. Such a method would, as the authors point out, be useful in that it (a) allows individual whiskers to be tracked over time among a field of many unlabeled whiskers, (b) could in principle allow this tracking to occur for multiple whiskers simultaneously with color-based image segmentation, (c) in principle allow higher luminance contrast compared with brightfield imaging. This is therefore a potentially important method. The authors actually do not directly show (a)-(c), but instead focus on testing the potential "deal-killer" downsides of the method, namely that it would significantly alter whisker mechanics and disrupt whisker behavior. They do a reasonable job of this testing and find that mechanics and behavior are not noticeably disrupted. There is room for improvement with respect to including information necessary for other labs to evaluate and apply the method. Detailed comments follow.

More information about the dye and its application is needed:

The authors list their dyes as simply "green and red dyes – Stargazer", but Stargazer has multiple green and red colors and it's not clear which exactly the authors used. Moreover, I could not find any product information online that would be useful for understanding its uses as a fluorescent marker of whiskers. What are the relevant chemical properties? (If Stargazer stops making these particular colors, or if one wanted to generalize the method, what factors would one use to choose a different dye?)

How much dye must be applied, for instance per unit of surface area or volume? The authors describe a timed application of 30 min. How much dye is deposited in this time? The authors could perhaps do a calibration based on fluorescence from a known volume of dye, or based on changes in weight. The amount of dye required for good signal-to-noise is important, for example in determining whether the method is likely to be useful for the much smaller whiskers of mice.

How quickly does the dye bleach? To what degree is fluorescent bleaching a limiting factor in use of the method?

Related to the efficiency of the fluorescence and how much dye must be applied, one would like to know how intense the illumination must be. Because bright light is a stressor for rodents and the illumination is in visible wavelengths, this is relevant for use in behavioral studies.

In the Introduction the authors mention a limitation of traditional (brightfield) whisker imaging, which they imply their method will overcome: "Unless the whisker is uniformly illuminated and reflects the light homogenously…image quality will be degraded." Uniform illumination (and detection) is also important with their method, however. This may be evident in their supplemental video: the fluorescence intensity appears to vary a lot from one frame to the next (e.g. frames 61 vs. 62). I would assume this could make comparing whisker shape from frame to frame more difficult. It's not clear that their method offers a way around the stringent illumination requirements of traditional approaches. Please discuss.

In the Introduction, the authors then state that a "higher degree of luminance contrast could solve many of the technical challenges inherent to videography." However, the authors do not report any quantification of the luminance contrast they achieve. They should probably add this.

In the Results: "Video recordings confirm the increased visibility of the dyed whiskers in the mobile rat, facilitating tracking (Supplementary video)." The authors don't actually demonstrate facilitated tracking. In fact, although they set up the paper in the Introduction as offering a useful alternative to traditional approaches to whisker imaging/tracking, the authors never actually demonstrate its use for tracking position and shape of whiskers on the intact animal. It seems important to show this.

In Figure 3, it would be helpful to show example maps of the dyed conditions for the same whiskers.

The assumption of a uniformly tapered conical shape for dyed whiskers should be justified with experimental measurements. In general, the authors should characterize the shape of a whisker before and after dye application. Although not as straightforward, I suggest that one way the authors could add significant value would be to estimate Young's modulus before and after dye application.

The authors claim their dye labeling method is "long-lasting" (subsection “Dye effects on mechanical properties”, last paragraph). Figure 4 show for cut whiskers the stability of mechanics across months (assuming "M" in the x-axis labels indicates months; please include in legend). However, I could not find information showing that the fluorescence is long-lasting. Please clarify.

*Reviewer #3:*

Rigosa et al. apply fluorescent dye to rat whiskers to enable high contrast visualisation of whisker motion. This method might be useful for many researchers.

The authors give very few details about the dye, how it is applied, how it is excited, and how the fluorescence is imaged. For a methods paper, this is rather surprising!

The authors do not appear to track fluorescent whisker movements in behaving rats. This would be an important proof of concept: to track fluorescent whiskers during task performance at 2 kHz.

I am not convinced that the overall advance is of sufficiently general importance for publication in *eLife*.

---

## [Author Response]

*[…] Essential changes:*

*1) Few details are given about the dye and how it is applied. Exactly what dye was used, as "Stargazer" sells multiple green and red colors.*

We have adapted the draft to include the requested information. Below is the relevant section of the revised manuscript in the section entitled “Application of fluorescent dye”.

“We used the “Hair Color Rinse Semi Permanent” line, in particular “UV Red” and “UV Green” in Figure 1. […] The specific cosmetic colorants for “UV Red” are: CI 42520, Basic Orange 31, Basic Violet 16, Basic Yellow 40, HC Yellow 4, CI 16255. The specific cosmetic colorants for “UV Green” are: Basic Yellow 40, CI 42090, CI 61570, HC Yellow 4, CI 62045.”

Also in the section “Application of fluorescent dye” we have added the following text:

“Semi-permanent hair rinse is designed to diffuse into and bind to the hair (Robbins 2001). In order to bleach the whiskers before dye application, we applied a gel of 3% hydrogen peroxide mixed with a booster powder (Robbins 2001, Table 4-3). The chemical bleaching process would be expected to follow a diffusion law, where the diffusivity for the human hair has been estimated as 1.8 x 10-9 cm2 /min (Robbins 2001, Figure 4-3). Afterwards, we washed the whiskers with distilled water, dried them, and then submerged them for 30 min in the hair color gel.”

The methodology described above is accompanied by a new figure in [Supplementary-material SD2-data].

*2) You need to demonstrate tracking position and shape of whiskers in the intact animal.*

To demonstrate tracking position and shape of whiskers in the intact animal, we video recorded naive rats as they explored a water spout (Video 2). [Supplementary-material SD1-data] is a frame of the video, now shot at 1,000 frame/sec.

For the whisker tracking we used an open source solution, ImageJ. We added the following text in the Results section “Dye-enhanced visualization”:

“We performed whisker tracking using JFilament (Smith et al. 2010), a plugin of ImageJ (Schneider et al. 2012), a public domain Java-based image processing program. JFilament can segment and track 2D/3D filaments and has been widely used for fluorescence microscopy images. It allows tracking of these elements through a sequence of frames and is thus suitable for whisker tracking.”

*3) Related to point (8), accurate measurements of deflection and bending require video recording at 1 kHz or more. Is there sufficient signal-to-noise for this to occur?*

As mentioned in the reply to point (8), we have performed video recording at 1,000 frame/sec. The signal-to-noise ratio at this frame rate remains sufficient for tracking. The tracked position of the dyed whisker has been added in an overlay to the new video, as described above.

*How much dye is deposited and what is the weight of the dye required for such a signal-to-noise ratio and how does this compare to the weight of the unpainted vibrissa? What is the intensity of the excitation light?*

Below, we reply first with the theoretical framework and then the precise empirical data required by the Editor. In the original draft, we thoroughly tested the frequency response function (FRF) of the vibrissae before and after the application of the fluorescent dye (Figure 4). The negligible before/after difference implies that the ratio between the Young modulus and the mass density was unchanged. For the FRF to be conserved, either the Young modulus and the mass density were not significantly altered (the simplest explanation) or else they both changed, but changed in a manner that kept their ratio constant (a more complex and thus less likely explanation). Assuming the same diffusion dynamics for the dye and the bleaching molecules, the penetration depth would be of the order of 1 μm, hence no significant change in whiskers’ mass would be expected. Beyond the theoretical framework, we carried out testing as described in Methods section “Quantification of the deposited dye and of the photobleaching.” The relevant text is:

“As an empirical verification, we measured the mass of a set of 34 whiskers (a multiwhisker sample reduces the likelihood of measurement error) by means of a scale (RADWAG AS 220.R2) with an accuracy of 0.1 mg. Prior to processing, the pooled mass was 14.8 mg. No difference in mass was detected after the chemical bleaching, while the coloring produced an increment of 0.1 mg (from 14.8 to 14.9 mg; within the measurement error of the scale), less than 0.7%.”

*4) Please comment on the bleaching rate of the dye. Please comment on the ability to re-stain the vibrissae.*

We measured the bleaching and modelled it with an exponential decay. The time constant under strong LED illumination is about 14 hours (about 50,000 seconds), equivalent to about 25,000 trials of 2 seconds ([Supplementary-material SD3-data]).

We added following text in Methods section “Quantification of the deposited dye and of the photobleaching”:

“We measured the photobleaching of the dye for seven hours with all 7 of the lamp’s LEDs pointing at the whiskers at a distance of about 15 cm. […] The measured time constant was about 14 hours (about 50,000 seconds), equivalent to about 25,000 trials of 2 seconds.”

*Reviewer #1:*

*[…] The authors examined four rats, well-trained in a whisker-mediated vibrotactile delayed comparison task (Fassihi 2014), for behavioral effects of dye application. However, this behavioral task does not depend on the mechanical properties of the whiskers, since whisker's movement is induced by mechanical deflection. I suggest taking these data out.*

The authors feel that the behavioral verification is critical to the work and the manuscript would be denatured by removal of these contents. At the cost of just two panels of one figure, we can give at least a substantial (if not complete) response to a question that many readers would likely pose: “The method improves whisker visibility, but how do rats deal with it?” We have to disagree with the reviewer’s contention that “this behavioral task does not depend on the mechanical properties of the whiskers, since whisker's movement is induced by mechanical deflection.” When mechanical deflection is applied to a guitar string, surely the properties of that string still matter. In the case of the selected behavioral task, transmission of mechanical energy from the contact point on the moving plate to the whisker follicle does depend on mechanical properties, as noted by reviewer #2.

*Reviewer #2:*

*Rigosa et al. offer a novel method of fluorescently dying rat whiskers in order to solve limitations of traditional high-speed video based on brightfield imaging. Such a method would, as the authors point out, be useful in that it (a) allows individual whiskers to be tracked over time among a field of many unlabeled whiskers, (b) could in principle allow this tracking to occur for multiple whiskers simultaneously with color-based image segmentation, (c) in principle allow higher luminance contrast compared with brightfield imaging. This is therefore a potentially important method. The authors actually do not directly show (a)-(c), but instead focus on testing the potential "deal-killer" downsides of the method, namely that it would significantly alter whisker mechanics and disrupt whisker behavior. They do a reasonable job of this testing and find that mechanics and behavior are not noticeably disrupted. There is room for improvement with respect to including information necessary for other labs to evaluate and apply the method.*

We thank the reviewer for the critique. In the revision, we have focused additional effort on issues (a)-(c), as requested by the reviewer. Some of changes detailed above for reviewer #1 are also relevant for reviewer #2; other changes are specific to reviewer #2 and are detailed below.

*Detailed comments follow.*

*More information about the dye and its application is needed:*

*The authors list their dyes as simply "green and red dyes – Stargazer", but Stargazer has multiple green and red colors and it's not clear which exactly the authors used. Moreover, I could not find any product information online that would be useful for understanding its uses as a fluorescent marker of whiskers. What are the relevant chemical properties? (If Stargazer stops making these particular colors, or if one wanted to generalize the method, what factors would one use to choose a different dye?)*

Our reply to this point was addressed directly to the Editor, above.

*How much dye must be applied, for instance per unit of surface area or volume? The authors describe a timed application of 30 min. How much dye is deposited in this time? The authors could perhaps do a calibration based on fluorescence from a known volume of dye, or based on changes in weight. The amount of dye required for good signal-to-noise is important, for example in determining whether the method is likely to be useful for the much smaller whiskers of mice.*

Likewise, our reply to this point was addressed directly to the Editor. In particular, we tested the mass difference and found it to be at most 1% (limited by scale accuracy). For this reason, it would not be a concern to increase the exposure time of the whisker to the dye in order to obtain higher contrast, nor to re-apply at different times.

*How quickly does the dye bleach? To what degree is fluorescent bleaching a limiting factor in use of the method?*

See reply to the Editor above.

*Related to the efficiency of the fluorescence and how much dye must be applied, one would like to know how intense the illumination must be. Because bright light is a stressor for rodents and the illumination is in visible wavelengths, this is relevant for use in behavioral studies.*

We successfully shot a video at 1,000 fps of a rat while behaving (drinking) and we provide whisker tracking for that video. We agree that in behavioral studies the light source should be positioned and angled in a manner that provides the highest possible illumination of whiskers, together with the lowest possible illumination everywhere else, especially in a manner that avoids direct shining into the eyes. But such configurations exceed the aim of this paper. In our experience, the rat does not avoid the area of the behavioral apparatus containing the light source. They show no aversive behavior or signs of stress (crouching, arching, excessive grooming, etc.). In the session where we shot the video, the rat ate and drank at the spout when it was in the illuminated position just as willingly as when the spout was in a dark corner. Last but not least, our veterinarian carried out an optical exam and found no detrimental effects. However, our paper is about whisker coloring and we believe that issues of illumination are beyond our scope.

*In the Introduction the authors mention a limitation of traditional (brightfield) whisker imaging, which they imply their method will overcome: "Unless the whisker is uniformly illuminated and reflects the light homogenously…image quality will be degraded." Uniform illumination (and detection) is also important with their method, however. This may be evident in their supplemental video: the fluorescence intensity appears to vary a lot from one frame to the next (e.g. frames 61 vs. 62). I would assume this could make comparing whisker shape from frame to frame more difficult. It's not clear that their method offers a way around the stringent illumination requirements of traditional approaches. Please discuss.*

Uniform illumination is crucial for our method as well. Motivated by this and the other reviewers, we put additional effort into the collection of high quality, well-illuminated video, which is now presented in the revision.

*In the Introduction, the authors then state that a "higher degree of luminance contrast could solve many of the technical challenges inherent to videography." However, the authors do not report any quantification of the luminance contrast they achieve. They should probably add this.*

We did not quantify the illumination because there is no fixed, agreed-upon level of illumination that will satisfy every investigator for every study. The statement is general and uniformly true: in every condition, a higher degree of luminance contrast is always better. We added a new video shot at 1,000 fps with appropriate light conditions and optical filtering. We provided information about the light conditions used for that video. We expect that investigators will compare the whisker to background contrast in that video, and in Figure 1, to their current methods and will appreciate the improved contrast.

*In the Results: "Video recordings confirm the increased visibility of the dyed whiskers in the mobile rat, facilitating tracking (supplementary video)." The authors don't actually demonstrate facilitated tracking. In fact, although they set up the paper in the Introduction as offering a useful alternative to traditional approaches to whisker imaging/tracking, the authors never actually demonstrate its use for tracking position and shape of whiskers on the intact animal. It seems important to show this.*

The revised manuscript demonstrates whisker tracking on a 1,000 fps video.

*In Figure 3, it would be helpful to show example maps of the dyed conditions for the same whiskers.*

Figure 3 only explains how to obtain the data that is shown in the matrices of Figure 4. Figure 4 fulfils the reviewer’s request.

*The assumption of a uniformly tapered conical shape for dyed whiskers should be justified with experimental measurements. In general, the authors should characterize the shape of a whisker before and after dye application. Although not as straightforward, I suggest that one way the authors could add significant value would be to estimate Young's modulus before and after dye application.*

We report no significant change in mass after the coloration (Methods section “Quantification of the deposited dye and of the photobleaching”) and no difference of mechanical dynamics, hence the direct conclusion is that Young modulus was not changed.

*The authors claim their dye labeling method is "long-lasting" (subsection “Dye effects on mechanical properties”, last paragraph). Figure 4 show for cut whiskers the stability of mechanics across months (assuming "M" in the x-axis labels indicates months; please include in legend). However, I could not find information showing that the fluorescence is long-lasting. Please clarify.*

We added a figure ([Supplementary-material SD3-data]) to detail photobleaching (i.e., how long the fluorophore continues to fluorescence after exposure to strong light). The lifetime, in terms of light exposure in an experimental apparatus, is longer than what would be needed in most studies.

*Reviewer #3:*

*Rigosa et al. apply fluorescent dye to rat whiskers to enable high contrast visualisation of whisker motion. This method might be useful for many researchers.*

*The authors give very few details about the dye, how it is applied, how it is excited, and how the fluorescence is imaged. For a methods paper, this is rather surprising!*

We have added this information to the revised manuscript.

*The authors do not appear to track fluorescent whisker movements in behaving rats. This would be an important proof of concept: to track fluorescent whiskers during task performance at 2 kHz.*

We added a video recorded at 1,000 fps. We demonstrated whisker tracking on that video. In the literature, 1 kHz is the most common frame rate used for tracking since whisker dynamics have been found not to exceed 500 Hz. In the same session we recorded data up to 2,000 fps but it did not seem to add much to the significance of paper.

*I am not convinced that the overall advance is of sufficiently general importance for publication in eLife.*

We regret hearing this, but we believe that this method would be helpful for many laboratories that study tactile behavior in rodents because it allows easy single-whisker visualization without trimming.